# Screening accuracy of a 14-day smartphone ambulatory assessment of depression symptoms and mood dynamics in a general population sample: Comparison with the PHQ-9 depression screening

**Sebastian Burchert** [1]*, **André Kerber**[1], **Johannes Zimmermann**[2], **Christine Knaevelsrud**[1]

**1** Division of Clinical Psychological Intervention, Department of Education and Psychology, Freie Universität Berlin, Berlin, Germany, **2** Institute of Psychology, University of Kassel, Kassel, Germany

* s.burchert@fu-berlin.de

## Abstract

### Introduction

Major depression affects over 300 million people worldwide, but cases are often detected late or remain undetected. This increases the risk of symptom deterioration and chronification. Consequently, there is a high demand for low threshold but clinically sound approaches to depression detection. Recent studies show a great willingness among users of mobile health apps to assess daily depression symptoms. In this pilot study, we present a provisional validation of the depression screening app Moodpath. The app offers a 14-day ambulatory assessment (AA) of depression symptoms based on the ICD-10 criteria as well as ecologically momentary mood ratings that allow the study of short-term mood dynamics.

### Materials and methods

N = 113 Moodpath users were selected through consecutive sampling and filled out the Patient Health Questionnaire (PHQ-9) after completing 14 days of AA with 3 question blocks (morning, midday, and evening) per day. The psychometric properties (sensitivity, specificity, accuracy) of the ambulatory Moodpath screening were assessed based on the retrospective PHQ-9 screening result. In addition, several indicators of mood dynamics (e.g. average, inertia, instability), were calculated and investigated for their individual and incremental predictive value using regression models.

### Results

We found a strong linear relationship between the PHQ-9 score and the AA Moodpath depression score (r = .76, p < .001). The app-based screening demonstrated a high sensitivity (.879) and acceptable specificity (.745). Different indicators of mood dynamics covered substantial amounts of PHQ-9 variance, depending on the number of days with mood data that were included in the analyses.

**Data Availability Statement:** The data underlying the results is available from the Zenodo database (DOI: 10.5281/zenodo.3384860).

**Funding:** Open Access Funding provided by the Freie Universität Berlin. The funders had no role in study design, data collection and analysis, decision to publish, or preparation of the manuscript.

**Competing interests:** The authors declare that the research was conducted in the absence of any commercial or financial relationships that could be construed as a potential conflict of interest. For the purpose of this study, Freie Universität Berlin and Aurora Health signed a formal data usage agreement. The data collection was conducted with regular users of the app. As part of the data usage agreement, Aurora Health provided the authors with a raw data export file from the app's database. This does not alter our adherence to PLOS ONE policies on sharing data and materials. Aurora Health was not involved in the preparation of the raw data, the data analysis, the writing of this article and the interpretation the results.

## Discussion

AA and PHQ-9 shared a large proportion of variance but may not measure exactly the same construct. This may be due to the differences in the underlying diagnostic systems or due to differences in momentary and retrospective assessments. Further validation through structured clinical interviews is indicated. The results suggest that ambulatory assessed mood indicators are a promising addition to multimodal depression screening tools. Improving app-based AA screenings requires adapted screening algorithms and corresponding methods for the analysis of dynamic processes over time.

## Introduction

Major depression affects over 300 million people worldwide. It has become one of the leading causes of loss of quality of life, work disability and premature mortality [1]. Effective, evidence-based therapeutic approaches are available but current estimates of the global treatment gap indicate that only between 7% (in low-income countries) and 28% (in high-income countries) of those with depression actually receive an intervention [2]. Structural and individual barriers lie at the root of this global public health issue. Healthcare systems often lack the financial and human resources to provide depression treatments at scale [3]. In addition, especially milder cases of major depression often remain undetected by primary care providers—the main and often only access point to care for the majority of patients with symptoms of depression [4]. Access to secondary care is often difficult due to factors such as high costs, regional unavailability or long waiting lists [5–7]. At the individual patient level, difficulties with identifying symptoms or the desire to handle emotional problems on one's own often occur in combination with a lack of trust in professionals or fear of stigmatization [6–8]. All of these factors reduce the chances of early detection, add to the issue of under-diagnosis and increase the risk of long-term symptom deterioration and chronification [9]. While general, population-wide screenings for depression remain a controversial topic in the literature [10, 11], there is an undeniable need to improve early recognition for those experiencing symptoms of depression as well as to provide low-threshold pathways to available mental health care systems.

Smartphone applications (apps) are increasingly being recognized as tools with the potential to provide scalable solutions for mental health self-monitoring, prevention and therapy support [12–14]. User-centred research shows that digital sources are perceived as a convenient and anonymous way of receiving a primary evaluation of symptoms [15]. Furthermore, the internet is currently the most frequented source of health information in general [16] and of mental health information in particular [17, 18]. Among internet users in a random sample of general practice patients in Oxfordshire (England), the willingness to seek mental health information online was found to be higher among persons who experience increased psychological distress and even higher in persons with a past history of mental health problems [17]. Smartphone apps combine the ease of access of online information with interactive elements and a seamless integration into everyday life. Consequently, mental health apps are already widely used among patients with symptoms of depression [19]. Initial studies found good response rates as well as a high willingness to complete smartphone-based symptom screenings [20–23]. Furthermore, a recent meta-analysis on randomized controlled trials found significant effects on depressive symptoms from smartphone-based mental health interventions [24]. Other authors report a positive impact on the willingness to seek out a professional about the

results of an app-based screening [25] and a positive effect on patient empowerment when combining self-monitoring with antidepressant treatment [26]. However, the number of freely available mental health apps has long become opaque for clinicians and end users seeking trustworthy, secure and evidence-based apps [21, 27, 28]. The mobile mental health market is rapidly expanding and there is a substantial gap between apps that were developed and tested in research trials and apps that are available to the public [29, 30]. Therefore, the promising early research findings on mental health apps do not necessarily apply to what the end users have access to. Several reviews on freely available mobile mental health apps have shown that the majority does not provide evidence-based content and was not scientifically tested for validity [29, 31, 32]. It is important to note that the willingness to use smartphone-based mental health services does not necessarily translate into actual use of such apps and that potential users often report concerns regarding the trustworthiness of apps in terms of their clinical benefit and privacy protection [23]. Furthermore, it becomes increasingly evident that mental health apps struggle with user engagement and retention [33]. These issues require further research on user-centred approaches to app development by focussing on the personal and contextual factors that shape usability requirements [34].

Nevertheless, smartphone-based approaches may have decisive advantages that underline the importance of scientific monitoring of the new technological developments in this field. One example is the comparison of smartphone-based symptom screenings with standard screening questionnaires. The latter are currently the most widely recommended assessment instruments for screening in primary and secondary care as well as in clinical-psychological research. Screening questionnaires such as the established Patient Health Questionnaire (PHQ-9) [35] usually require respondents to retrospectively report symptoms that occurred over a longer period of time (e.g. over the past two weeks). For this reason, such instruments are susceptible to a number of potential distortions caused by recall effects, current mood, unrelated physiological states (e.g. pain), fatigue or other situational factors at the time of assessment [12, 36–38]. Smartphones, however, provide the technical capabilities to gather more ecologically valid data through ambulatory assessment (AA) [25, 39]. In AA, emotions, cognitions or behaviour of individuals are repeatedly assessed in their natural environments [40]. For example, instead of asking respondents to retrospectively judge how often they have been bothered by feeling hopeless over the past two weeks, participants are prompted to answer sets of short questions on current feelings, several times a day for a period of two weeks. As summarized by Trull and Ebner-Priemer [41], distinct advantages of AA are the ability to gather longitudinal data with high ecological validity as well as the ability to investigate dynamic processes through short assessment intervals that would be difficult to assess retrospectively. Common limitations are the technical requirements, compliance with AA study protocols–e.g. due to repetitiveness, long assessments, or frequent reminders–as well as privacy concerns [41]. Still, the AA methodology is gaining increasing attention in the field of psychological assessment [42–44] with initial studies indicating comparable or even better performance of AA measures in direct comparison with standard paper-pencil measures [45, 46].

Despite the promising early findings on app-based screenings, there is still a lack of instruments that were specifically designed and validated for use in AA on smartphones. Therefore, AA studies often make use of established paper-pencil questionnaires such as the PHQ-9. This approach to AA introduces new challenges, especially in regard to user engagement—one of the most pressing issues in the mental health app field [47]. The repeated assessment of questionnaire batteries is time consuming and potentially repetitive for participants which may lead to participant fatigue and a quick drop in the number of users who continue using the app. In recent years, several studies reported severe problems with adherence that significantly undermine the advantage of scalability in smartphone-based mental health approaches [48–

51]. In research settings this frequently required additional (i.e. often monetary) incentives [22] which will not be available in real-world dissemination. On the contrary, a strong focus on usability and attention to user-centred design is of great importance when designing apps that keep users engaged [34, 47, 52]. More user-centred research is required to develop smartphone-based depression screening apps that balance psychometric properties, depth of data and user engagement to reach feasible and clinically-sound tools for use outside of the research setting.

This also includes to look beyond established tools and to investigate new opportunities that smartphone-based AA opens to study subtler correlates of depression based on very short but frequent assessments. Factors such as short-term mood dynamics are associated with psychological wellbeing in general [53] and depression in particular [54]. Self-monitored momentary mood ratings were found to be significantly correlated with clinical depression rating scales [46, 55, 56]. Moreover, momentary affect states were found to fluctuate in depressed patients, even in those with severe depression [54]. Within-person dynamics in mood states can be characterised by high or low difference of successive mood ratings (i.e. mood instability), high or low overall variability of mood ratings, and high or low autocorrelation of successive mood ratings (i.e. inertia) [53, 57, 58]. While high variability of mood indicates a large range of experienced mood states, high mood instability indicates frequent moment-to-moment fluctuations in the sequence of mood states. Inertia, on the other hand, is a parameter for the temporal dependency of subsequent mood states, with high inertia of mood states indicating a greater resistance to affective change [59]. Research on the relation between mood instability, variability or inertia and depression is inconclusive regarding the additional value of these mood indicators compared to average mood. Consequently, these variables are not yet part of established depression screening approaches. For example, in their paper on affect dynamics in relation to depressive symptoms, Koval et al. [57] discuss a number of different indices for mood dynamics. They replicate initial findings that people with depression symptoms show greater mood instability and variability as well as higher mood inertia than non-depressed people. However, amongst all measures of affect dynamics, average levels of negative affect were found to have the highest predictive validity regarding symptoms of depression. This finding was recently replicated by Bos et al. [58].

AA is a feasible approach to repeatedly assess mood ratings over a longer period which is the basis for the calculation of mood indices. Progress in this field of research has the potential to translate into improved mental health screening algorithms that go beyond the standard assessments. In this context, it has also been pointed out that combinations of multiple measurements on the basis of mobile technology may improve diagnostic accuracy and the prediction of treatment trajectories in the mental health field [60, 61], an approach that is already far more common in medical diagnostics [62]. Early detection could profit as well, for example through the identification of indicators of critical slowing down in mood dynamics as an approach to identify points at which individuals transitions into depression [63, 64]. Furthermore, there is indication that self-monitoring of mood can have positive effects on emotional self-awareness [65]. However, it also needs to be considered that self-tracking may have negative effects [66–68]. While the evidence base on this is still very limited, it is still recommended in the literature to balance the burden for users with the predictive value of the data [69].

In this pilot study we describe a preliminary evaluation of the depression screening component of the freely available mental health app Moodpath (Aurora Health GmbH, 2017) that utilizes AA to assess symptoms of depression as well as mood in daily life. Our study had two main aims: (1) To conduct a preliminary psychometric validation of the Moodpath depression screening by comparing it with the established PHQ-9 screening questionnaire and (2) to conduct exploratory analyses on indicators of mood dynamics to gain knowledge on their

potential incremental value in the detection of depression. In contrast to previous studies with similar methodologies, this study puts the emphasis on an extended 14-day ambulatory assessment of depression symptoms and momentary mood in combination with the retrospective assessment of depression symptoms for the exact same 14 days.

## Materials and methods

### Recruitment and participants

A convenience sample of N = 200 participants was selected through consecutive sampling among users of the German language iOS version of Moodpath. The sample size was chosen based on the estimated number of app users to complete the 14-day assessment within a period of approximately one month. There were no inclusion or exclusion criteria but agreement to the general terms and conditions of using the app and consent to participation in the additional PHQ-9 assessment. The general terms included agreement to the completely anonymous use of app data for scientific research purposes. Moodpath users come from the general population and learn about the app through web- or app store search, social media ads or media coverage. Participants in this study were regular users of the app who had started using the app before being asked to participate in the study. All users who completed the app's regular 14-day screening phase were contacted automatically through an in-app notification until the sample size of 200 was reached (see Fig 2). The recruitment took place in January 2017. No additional recruiting procedures were used. The in-app notification was sent after 14 days and provided further information on the purpose of the additional assessment and on the fully anonymous and voluntary nature of the additional questions. Participants provided electronic informed consent and were free to decline answering any or all the additional questions without any consequences for their regular use of the app. The study procedures were approved by the Freie Universität Berlin Institutional Review Board.

### Assessment instruments

**Moodpath depression screening.** Moodpath was developed by the Berlin-based startup company Aurora Health GmbH and is one of the first German- and English-language solutions for depression screening specifically developed for smartphones. The Moodpath depression screening focuses on an anonymous, user-friendly and low-burden ambulatory assessment of depression symptoms in order to optimize adherence and applies the diagnostic principles of the ICD-10 classification [70]. Three daily assessments with 3 questions and one mood rating each are collected over a period of 14 days and beyond. The app is a certified medical product (CE) and available free of charge for Android and iOS. Users of the app received no monetary incentives to use the app. However, participants received feedback on their symptoms in the form of a detailed summary report after completing the 14-day assessment period. In this study, Moodpath release version 2.0.7 was used.

The Moodpath app utilizes a set of 45 questions that were developed for use in ambulatory assessment within this app (S1 Table). 17 of these questions (Table 1) cover all 10 symptoms of depression described in the ICD-10 [70]. Except for suicidal thoughts, diminished appetite and disturbed sleep, all symptoms are covered by two different items representing facets of the respective symptom. The remaining 28 questions assess additional somatic symptoms, other mental health issues, resources, and wellbeing. These items were added to the Moodpath screening to reduce repetitiveness of the assessment and to identify additional symptoms as well as resources for further exploration by the user or a therapist with access to the data. The additional items did not cover ICD-10 symptoms of depression and most were not assessed repeatedly. Therefore, the additional items are not part of the analyses presented in this paper.

**Table 1. Moodpath depression screening questions and ICD-10 symptom categories.**

| ICD-10 symptoms | | Moodpath questions |
|---|---|---|
| **Core symptoms of depression** | | |
| 1 | Depressed mood | Are you feeling depressed? |
| | | Are you feeling hopeless? |
| 2 | Loss of interest and enjoyment | Do you feel like you are not interested in anything right now? |
| | | Do you have less pleasure in doing things you usually enjoy? |
| 3 | Increased fatigability | Do you currently have considerably less energy? |
| | | Are your everyday tasks making you very tired currently? |
| **Associated symptoms of depression** | | |
| 4 | Reduced concentration and attention | Is it hard for you to make decisions currently? |
| | | Is it hard for you to concentrate currently? |
| 5 | Reduced self-esteem and self-confidence | Is your self-confidence clearly lower than usual? |
| | | Are you feeling up to your tasks? |
| 6 | Ideas of guilt and unworthiness | Are you blaming yourself currently? |
| | | Do you think you are worth less than others right now? |
| 7 | Bleak and pessimistic views of the future | Are you thinking that you will be doing well in the future? |
| | | Are you looking hopefully into the future? |
| 8 | Ideas or acts of self-harm or suicide | Are you thinking about death more often than usual? |
| 9 | Disturbed sleep | Did you sleep badly last night? |
| 10 | Diminished appetite | Do you have less or no appetite today? |

The answer format is "yes/no". If "yes" is chosen, the symptom burden is rated on a four-point scale.

A 14-day screening period was chosen to comply with the ICD-10 definition of depression. Users were prompted to complete 3 daily assessments (morning, midday, and evening). At each assessment, a block of 3 non-randomly selected questions was asked and had to be answered within a time window of 5 hours. Otherwise, the set was handled as missing data. The order and content of the question blocks was predefined and identical for each participant (S2 Table). Blocks were divided into obligatory blocks and optional blocks. All obligatory blocks combined contained the minimum of questions required to generate the 14-day report (see below). If an obligatory block was missed, it automatically took the place of the same block on the next day (e.g. next day's morning block if a morning block was missed). Per time window, participants received one automatic push notification that reminded them to answer the questions. The assessment progress was visualized in the form of a path that also functioned as the user interface (Fig 1) providing immediate feedback on the availability of new questions, completeness and progress towards the 14-day summary at the end of the assessment period.

To generate the 14-day report, a minimum of 4 assessment points for each depression symptom (Table 1) is required. This applies for all symptoms except for item 8 (ideas of self-harm or suicide) that has a minimum requirement of two assessments. The exception was introduced based on user feedback indicating that this question was too common. For symptoms covered by two different items, the requirement is that each of these symptoms must be assessed at least twice. Consequently, not all symptoms and questions are assessed at every assessment point or assessment day. This reduces repetitiveness and the overall burden for users at each individual assessment point. In order to still ensure enough data points for each symptom, the differentiation into obligatory and optional blocks was introduced. This system ensured that missed blocks containing questions on depression symptoms for which a minimum of 4 assessments was not reached yet were repeated. Participants answer each question,

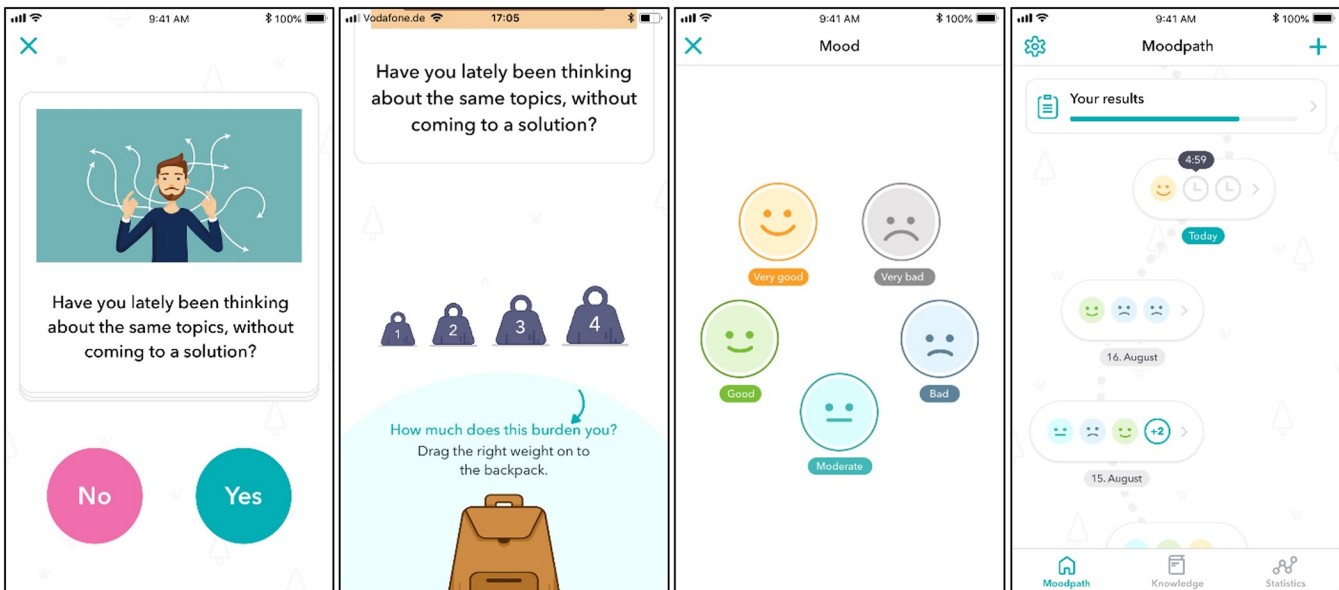

**Fig 1. Moodpath screens (images reproduced with permission from Aurora Health GmbH).**

e.g. "Are you feeling depressed?", on a dichotomous scale with "Yes" or "No". If a symptom is confirmed, the severity ("How much does this affect or burden you?") is rated on a four-point scale ranging from 1 to 4 with a visual anchor (Fig 1). In the absence of an established cut-off at the individual item level, we defined a preliminary cut-off and considered symptom ratings with a severity of 2 or higher as potentially clinically significant.

Based on the symptom rating, the Moodpath depression score is calculated as the sum of all depression symptoms that occurred with clinical significance on more than half of the ambulatory assessments (i.e. 3 or 4 in case of the 4 minimum assessments). This rule is based on the PHQ-9 algorithm method in which only items rated as at least 2 (more than half the days) are counted towards the symptom score [71]. In accordance with the PHQ-9 algorithm method, an exception was implemented for item 8 (ideas of self-harm or suicide). This item is counted if it occurred with clinical significance on at least half of the ambulatory assessments. Consequently, the score can vary between 0 and 10, with 0 = no depression symptom was reported with a severity of 2 or more on more than half of the (at least 4) different assessment points for that symptom. The same rule applies to symptoms covered by two different items. In case of 4 assessments of the symptom, one item had to be above the cut-off at least once while the other item had to be above the cut-off at least twice. Users of the app receive a report on the severity of their symptoms that takes into account the Moodpath depression score as well as the ICD-10 criteria for depression [70]: i.e. 2 out of 3 core symptoms + 2 out of 7 associated symptoms were reported as indication of "mild depression", 2 core + 3 or 4 associated symptoms were reported as "moderate depression" and 3 core + 5 or more associated symptoms were reported as potential "severe depression". With the report, users received the following disclaimer: "The Moodpath App screening is not a substitute for a medical diagnosis. It should only be used as an indication to seek professional consultation". For the purpose of validation, in this study the above described ICD-10 criteria for "moderate depression" (i.e. 2 + 3) were used as indication of clinically relevant symptoms.

**Mood tracking.** At each assessment point, participants were presented with a single item bipolar mood rating scale asking how they are currently feeling. The scale consists of 5 smiley-

faces ranging from sad over neutral to happy (Fig 1). During the assessment, the app converted the selected smileys to numerical values ranging from 0 to 4, with 0 indicating negative mood.

**PHQ-9.** For the initial validation of the Moodpath screening, the established PHQ-9 questionnaire with 9 questions as published by Kroenke et al. [35] was chosen in its German translation [72] as reference criterion due to strong evidence on its screening utility [73, 74]. The additional PHQ-9 question on functioning [35] was not asked. Due to resource limitations, it was not possible to conduct clinical interviews at this stage of the evaluation. However, the PHQ-9 is commonly used to calculate a sum score with cut-off scores to identify minimal (1–4), mild (5–9), moderate (10–14), moderately severe (15–19) and severe (20–27) symptom severity [35]. In addition, an algorithm can be used to approximate a provisional diagnosis on the basis of the DSM-IV diagnostic criteria [71]–which are (with the exception of an adjustment to the bereavement exclusion criterion) identical to the DSM-5 criteria. For this, 5 or more out of 9 symptoms need to be given for most of the day, on nearly every day during a period of 14 days. These symptoms further need to include either depressed mood or little interest or pleasure in doing things [71]. A score of 2 ("more than half the days") or higher on an item is treated as clinically significant on all items except item 9 (i.e. thoughts of death or hurting oneself). For the latter, a score of 1 ("several days") is considered as clinically significant.

## Statistical analyses

**Psychometric evaluation of the depression screening.** All analyses were conducted in R [75]. Pearson product-moment correlation analyses were conducted to assess the statistical associations between the Moodpath depression score, the PHQ-9 score and participant age. Gender differences were assessed with two-sided t-tests and Cohen's d effect size calculations. The sensitivity of the ICD-based Moodpath screening criterion of 2 core + 3 associated symptoms was calculated as the percentage of participants correctly classified as potentially depressed out of all participants with a PHQ-9 categorical result above the cut-off of 5 symptoms. The specificity of the Moodpath screening was calculated as the percentage of participants correctly classified as not depressed out of all participants with a PHQ-9 result below the clinical cut-off.

In addition, exploratory methods were applied to identify the ideal Moodpath sum score cut-off without taking into account ICD-10 criteria for core and associated symptoms. To this end, sensitivity, and 1-specificity for different Moodpath depression scores were plotted as a receiver operating characteristic (ROC) curve, which provides the area under the curve (AUC) as a measure of accuracy. 95% confidence intervals for AUC, sensitivity and specificity were calculated with the pROC package for R [76]. Guidelines for interpreting AUC values suggest values above .80 as indication of good accuracy and values above .90 as indication of excellent accuracy [77]. The optimal cut-off value for the Moodpath depression score was calculated based on the sum of the squared distances from a 100% true positive rate (TPR) and a 0% false positive rate (FPR). This approach applied equal weights to sensitivity and specificity [78].

**Exploratory analyses on mood dynamics.** To test for a general (linear) pattern of mood development over time, we calculated a linear mixed effects model with mood ratings as dependent variable and time as independent variable. Time was defined as the number of days prior to answering the PHQ-9 while day was defined as a 24-hour time window. Consequently, day one is defined as the 24 hours prior to answering the PHQ-9. A linear mixed effects model seems appropriate because the data has a nested structure, with mood ratings (Level 1) nested in participants (Level 2). The model included a random intercept capturing interindividual differences in the average level of mood and a random slope capturing interindividual differences in mood change. We used Satterthwaite's approximations to derive p-values for fixed effects. To differentiate the effect of time and the effect of tracking frequency (i.e., number of

assessments), we calculated a second model with time, number of assessments as well as their interaction as fixed effects.

Furthermore, we calculated several statistics based on the ambulatory assessment of mood data to analyze associations between indicators of mood dynamics and the other study variables. Following recommendations by Koval et al. [57], we calculated the mood average as the mean over the 14-day assessment period up to the PHQ-9 assessment and additional indicators of mood dynamics. Mood instability was calculated as the root mean square of successive differences of subsequent mood ratings (mood RMSSD) and mood variability was calculated as the within-participant overall standard deviation of the mood ratings (mood SD). Mood inertia was calculated as the autocorrelation of subsequent within-participant mood ratings (mood autocorrelation). Finally, we calculated mood minimum (i.e. maximally negative mood rating) and mood maximum (i.e. maximally positive mood rating) for each participant. To analyze the temporal pattern of associations between these mood-based statistics and the PHQ-9 score, we separately calculated mood average, mood SD, mood RMSSD, mood autocorrelation, mood maximum and mood minimum based on one up to 14 days of mood data up to the PHQ-9 assessment. On this basis, we were able to calculate 14 separate regression models per mood statistic, each predicting the PHQ-9 score. Each of these 14 models differed in the number of days with mood data that were used for the calculations. While model one only used data of the day immediately up to the PHQ-9 assessment, model 14 used the aggregated mood data of all 14 days up to PHQ-9 assessment. This allowed further explorative analyses on the influence of the duration of ambulatory data collection on the predictive validity of the calculated indicators of mood dynamics. Finally, to estimate the effect of momentary mood on the PHQ-9 score, we performed a multiple linear regression analysis with the Moodpath depression score and the mood average within the 24-hour time window prior to the PHQ-9 assessment as predictors. The main research interest behind these exploratory analyses was to obtain information on how the duration of the ambulatory assessment affects the predictive validity of the data. To account for the problem of multiple testing, we adjusted the significance level to $p < .01$.

## Results

### Sample

Of the 200 selected users, N = 178 agreed to participate and answered the additional questions (Fig 2). These users generated a total of 4973 ambulatory assessments. On first inspection of the dataset, N = 65 cases had to be excluded due to an insufficient AA completion rate within the 14-day assessment period. The main reasons for this were that the app version used for this study accepted user input within a period of up to 17 days in cases of missing values. In addition, some participants answered the PHQ-9 with a latency of one or more days after completing the Moodpath screening. This also led to non-matching assessment periods. To ensure that all measures covered the exact same period, it was decided to exclude these cases. Consequently, the final dataset for all analyses in this paper was reduced to N = 113 cases. Average scores on all main study variables within the excluded subgroup did not differ significantly from the remaining sample (all p-values > .05). The average age of the final sample was 28.56 (SD = 9.54) with N = 84 (74%) of the participants being female, N = 21 (19%) male and N = 8 (7%) missing values for the gender variable.

### Descriptive statistics

The average number of ambulatory assessments (AAs) per included person was 27.94 (SD = 11.09) out of 42 with a skew of -0.55. Consequently, about two thirds of the AAs were

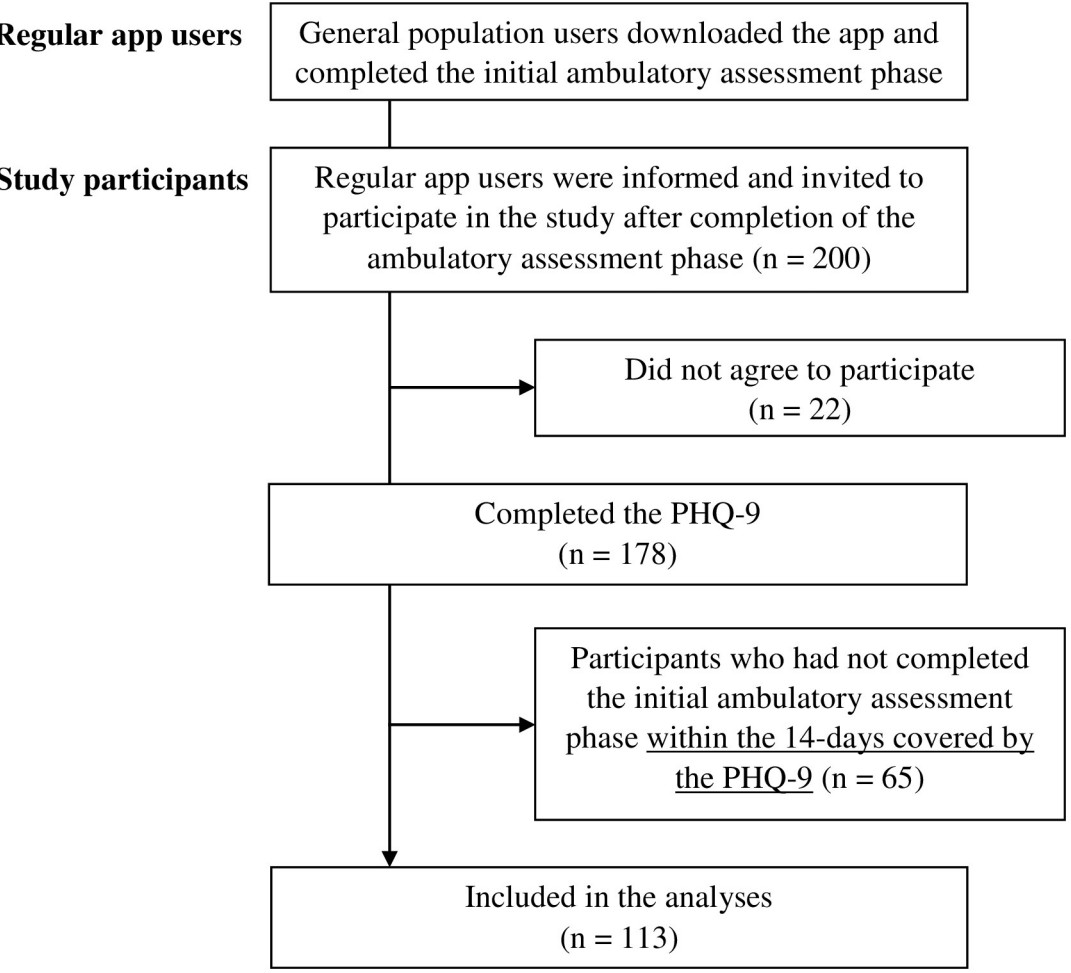

**Fig 2. Participant flow diagram.**

completed, indicating that the majority of the participants responded regularly and at least two times a day. The distribution of data points over the day shows a slightly higher number of responses in the evening. This is also reflected by how the percentages of missing data of the AA ratings were distributed: There were 20.4% missing data in the morning, 19.7% in the mid-day period and 9.1% missing data in the evening period, suggesting that evening assessments were less likely to be missed.

The average PHQ-9 score was 15.62 (SD = 5.96), indicating that the majority of the participants showed symptoms of moderately severe depression [35]. The PHQ-9 algorithm approach based on DSM-5 criteria for Major Depression revealed that 58.4% (n = 66) of the participants reported symptoms suggesting a potential episode of major depression. Based on the PHQ-9 severity cut-off scores, participants were further grouped into symptom severity levels (Table 2). The average 14-day Moodpath depression score was 5.67 (SD = 3.22). Based on the ICD-10 criterion for moderate depression, Moodpath identified 60.2% (n = 68) of the participants as potential cases (Table 3). Table 2 depicts ICD-10 based Moodpath depression severity levels. The average mood score was 1.94 (SD = 0.89), indicating a tendency towards experiencing negative mood states. Participant age was not correlated with any of the study variables. However, women had slightly higher 14-day Moodpath depression scores than men

**Table 2. Symptom severity levels.**

| | Symptom severity levels | | | | |
|---|---|---|---|---|---|
| | Minimal or none (n) | Mild (n) | Moderate (n) | Moderately severe (n) | Severe (n) |
| PHQ-9 | 2 | 21 | 26 | 31 | 33 |
| Moodpath | 42 | 3 | 28 | / | 40 |

PHQ-9 severity levels are based on sum score cut-offs; Moodpath severity levels are based on ICD-10 symptom count criteria.

with $M_{female}$ = 6.08 (SD = 3.23) and $M_{male}$ = 4.38 (SD = 2.77), t(35) = 2.44, p = .020, with a medium effect size of Cohen's d = .57.

## Comparison with the PHQ-9 screening

The results indicate a strong linear relationship between the retrospectively assessed PHQ-9 score and the aggregated momentarily assessed Moodpath depression score, r = .76, p < .001 (Table 4). The sensitivity and specificity of the Moodpath screening with the provisional cut-off at 5 and taking into account ICD-10 core and associated symptoms (i.e. at least 2 core + at least 3 associated symptoms) were .879, 95% CI [.775, .946] and .787, 95% CI [.643, .893], respectively. In other words, 58 out of 66 cases with potential depression and 37 out of 47 cases without potential depression (both according to the PHQ-9 screening) were identified correctly based on the Moodpath depression score, taking into account the ICD-10 criteria for moderate depression. In addition, Table 5 shows the sensitivity and specificity at different cut-off scores for the Moodpath depression score without differentiating between core and associated symptoms. Fig 3 shows the ROC curve of all potential cut-off points of the Moodpath depression score for potential depression based on the PHQ-9 result. The AUC was .880, 95% CI [.812, .948] indicating a good overall screening accuracy. The optimum cut-off based on equal weights to sensitivity and specificity confirmed the provisional cut-point at a sum score of 5 (Table 5). This indicates a very minor and—given the confidence intervals statistically non-significant—difference in sensitivity if core and associated symptoms are considered in addition to a sum-based cut-off point.

## Mood dynamics

Results of the linear mixed effects model with mood ratings as dependent variable showed an overall average decrease of 0.0126 (SD = 0.005) in mood ratings per day over the whole assessment period of 14 days (i.e. a total mean decrease by 0.177 over 14 days), t(164) = -2.77, p = .006. To calculate an effect size, we divided the 14-day total mean decrease by the standard deviation of the sample's mood ratings on the first day of the ambulatory assessment (SD = 0.89). This resulted in an effect size of d = -0.198 for the decrease in mood ratings over time. In a second analysis with time, number of assessments and their interaction as fixed effects, the number of assessments and the interaction were not associated with mood ratings.

**Table 3. Cross tabulation of the index test results.**

| | Moodpath cases | | |
|---|---|---|---|
| PHQ-9 cases | - (n) | + (n) | Total (n) |
| - | 37 | 10 | 47 |
| + | 8 | 58 | 66 |
| Total | 45 | 68 | 113 |

PHQ-9 cases are based on DSM-5 MDD criteria; Moodpath cases are based on ICD-10 criteria for moderate MDD.

**Table 4. Correlations between PHQ-9 and Moodpath depression scores (N = 113).**

| Variable | M | SD | MP score | Mood average | Mood SD | Mood RMSSD | Mood max | Mood min | Mood autocorr | Age |
|---|---|---|---|---|---|---|---|---|---|---|
| **PHQ-9 score** | 15.62 | 5.96 | .76 *** | -.66 *** | .31 *** | .29** | -.29** | -.59*** | .03 | -.09 |
| **MP score** | 5.67 | 3.22 | - | -.82 *** | .20* | .12 | -.50*** | -.61*** | .20* | -.04 |
| **Mood average** | 1.86 | 0.99 | | - | -.20* | -.08 | .65*** | .67*** | -.18 | .14 |
| **Mood SD** | 0.56 | 0.30 | | | - | .86*** | .41*** | -.53*** | .19 | -.15 |
| **Mood RMSSD** | 0.86 | 0.27 | | | | - | .45*** | -.48*** | -.25** | -.12 |
| **Mood max** | 3.09 | 0.71 | | | | | - | .27** | -.07 | .04 |
| **Mood min** | 0.50 | 0.70 | | | | | | - | -.17 | .18 |
| **Mood autocorr** | 0.24 | 0.21 | | | | | | | - | -.08 |

MP score = Moodpath depression score; Mood = Ambulatory assessment of momentary mood; Mood SD = mood variability; Mood RMSSD = mood instability; Mood autocorrelation = mood inertia.

* p < .05

** p < .01

*** p < .001.

A clear association between negative mood states and depression symptoms was found. A lower 14-day mood average was strongly associated with higher PHQ-9 scores (r = -.66, p < .001) as well as with higher Moodpath depression scores (r = -.82, p < .001). As depicted in Table 4, 14-day mood SD and RMSSD were weakly associated with higher PHQ-9 scores. Both, 14-day mood minimum and mood maximum were negatively correlated with PHQ-9 and Moodpath depression scores. However, while lower 14-day mood minimum was a strong predictor for higher PHQ-9 scores, 14-day mood maximum was only weakly associated with PHQ-9 scores. Despite having weak associations with lower mood average, higher mood SD and lower mood RMSSD, mood autocorrelation was neither associated with the PHQ-9 scores nor with the Moodpath depression score. Furthermore, all mood statistics except mood RMSSD were weakly to strongly correlated with the 14-day mood average.

Linear regression analysis revealed that 14-day mood average predicted a significant amount of variance of the PHQ-9 scores, $F(1, 111) = 85.39$, $p < .001$, $R^2 = .43$. The explained variance for this analysis and analyses with less than 14 days of data is displayed in Fig 4 (line 6). In a multiple linear regression analysis, mood RMSSD across 14 days predicted additional variance on top of mood average, $\Delta R^2 = .06$, $p < .001$ (line 4). This was also the case for 14-day mood SD, $\Delta R^2 = .04$, $p = .008$ (line 5) but not for mood autocorrelation, when added in a stepwise regression model on top of mood average. The Moodpath depression score alone covered a significant amount of the variance in the PHQ ratings, $F(1,111) = 152.9$, $p < .001$, $R^2 = .58$

**Table 5. Sensitivity and specificity at different cut-offs for the Moodpath depression score.**

| Moodpath cut-off | Sensitivity; [95% CI] | Specificity; [95% CI] |
|---|---|---|
| ≥ 9 | .42; [.30, .55] | .96; [.89, 1.00] |
| ≥ 8 | .58; [.45, .70] | .85; [.74, .93] |
| ≥ 7 | .71; [.61, .82] | .85; [.74, .93] |
| ≥ 6 | .79; [.68, .88] | .79; [.66, .89] |
| ≥ 5 | .91; [.83, .97] | .74; [.62, .85] |
| ≥ 4 | .95; [.89, 1.00] | .66; [.53, .78] |
| ≥ 3 | 1.00; [1.00, 1.00] | .51; [.38, .66] |
| ≥ 2 | 1.00; [1.00, 1.00] | .36; [.23, .50] |
| ≥ 1 | 1.00; [1.00, 1.00] | .19; [.09, .30] |

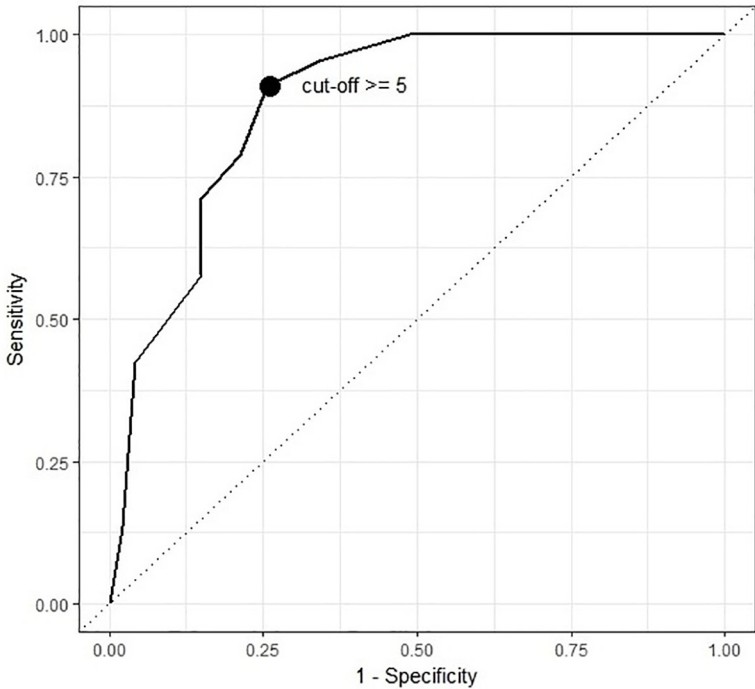

**Fig 3. ROC curve of Moodpath depression score and PHQ-9 score.**

(line 3). Entering the 14-day mood average or mood autocorrelation into the regression after the Moodpath depression score did not add explained variance. However, 14-day mood RMSSD added a significant amount of explained variance on top of the Moodpath depression score, $\Delta R^2$ = .04, p < .001 (line 1). This was also the case for mood SD, $\Delta R^2$ = .03, p = .006 (line

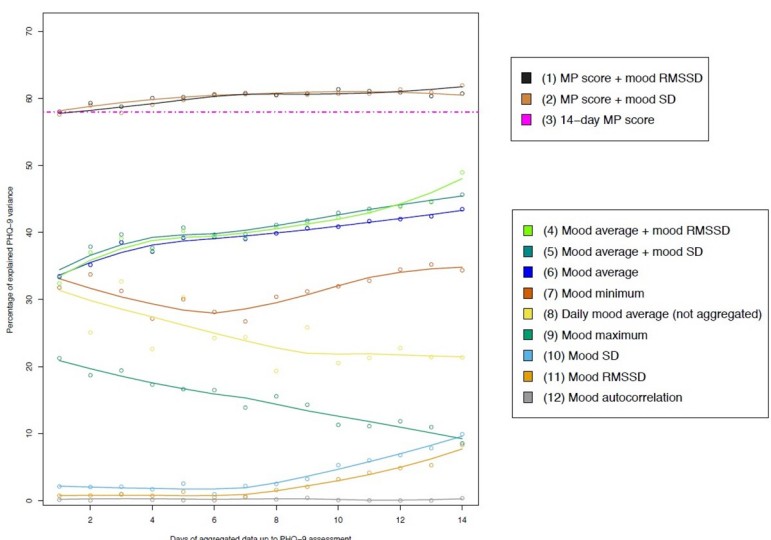

**Fig 4. PHQ-9 variance explained in multiple linear regressions calculated with data from one up to 14 days using different indicators of mood dynamics, ambulatory assessed depression symptoms and their combinations.** MP score = Moodpath depression score; Mood = Ambulatory assessment of momentary mood; Mood SD = mood variability; Mood RMSSD = mood instability; Mood autocorrelation = mood inertia.

2). Due to medium to strong intercorrelations (Table 4), no multivariate models with the Moodpath depression score and mood average, mood minimum or mood maximum were calculated to prevent multicollinearity.

Considering the incremental information of the ambulatory dataset over time, we calculated regression models with aggregated mood ratings from one up to 14 days up to the PHQ-9 assessment (Fig 4). This revealed that the explained variance of the PHQ-9 score was higher the more days were used for the calculation of the mood average (Fig 4, line 6). Regression models with mood SD (line 10) or mood RMSSD (line 11) as predictors also resulted in more explained variance with more days of available data. However, explained variance was low with a range between 2% and 9%. In comparison, mood minimum (line 7) explained a relatively stable 28% to 32% of the PHQ-9 variance and was less affected by the number of days available for the calculations. A divergent pattern was found for mood maximum: Here the explained variance for the prediction of PHQ-9 score was highest if only the days directly before the PHQ-9 assessment were taken into account for the calculation. Regression analyses with the daily mood average as the predictor (line 8) revealed a similar pattern. Compared to the aggregated mood average, the day-specific mood average immediately prior to PHQ-9 explained more variance than day-specific mood averages with greater temporal distance. In a multiple linear regression model, mood average of the 24 hours immediately prior to PHQ-9 assessment explained additional PHQ-9 variance on top of the Moodpath depression score, $\Delta R^2 = .03$, p = .006. Mood autocorrelation (line 12) was no predictor for depressive symptoms, irrespective of the duration of assessment. Associations between PHQ-9 score and mood average + mood RMSSD (line 4) showed an increase in explained variance with an increase in available days of data. This was also the case for mood average in combination with mood SD (line 5) as predictors.

All indicators of mood dynamics were compared between different PHQ-9 depression severity groups. Statistically significant differences were found for mood RMSSD between the mild and moderately severe symptom groups, t(45) = -3.26, p = .002 and the mild and severe symptoms groups, t(45) = -2. 90, p = .006 as well as for mood SD between the mild and severe symptoms groups t(48) = -3.10, p = .003. S1–S4 Figs contain additional figures that provide further information on group differences regarding the other indicators of mood dynamics.

## Discussion

This study is one of the first to compare a smartphone-optimized depression screening algorithm with an established screening questionnaire. This paper provides preliminary evidence on the Moodpath depression screening and the potential incremental value of momentary mood tracking over a period of 14 days. Overall, the initial results are promising. We found a strong positive association between the 14-day Moodpath screening and the retrospective PHQ-9 questionnaire but also a large amount of variance that was not shared between both measures. Preliminary results on the good screening accuracy of the app further underline the importance of validating the tool. The exploratory findings on indicators of mood dynamics indicate that they are an interesting data source with relevance for more elaborate, multimodal screening algorithms. But before discussing the findings of the study in further detail, several important limitations need to be pointed out.

### Limitations

Firstly, this study was intended as a pilot to estimate the potential of a more resource intensive validation of the tool. The pilot was designed to interfere as little as possible with the overall user experience (UX) of the app. UX is defined in ISO 9241–210 [79] as „a person's

perceptions and responses that result from the use and/or anticipated use of a product, system or service.". Good UX is crucial in ensuring user motivation to continue using the app and can be negatively affected by extensive data collection. Therefore, the recruitment period was limited to ensure that only a small part of the Moodpath user base was asked to participate. In addition, the Moodpath screening was compared to a short and established screening questionnaire but not to a clinical interview–the gold standard in clinical assessment studies. Here as well, the aim was to minimize the potentially negative effects of lengthy assessments on the UX. Questionnaires such as the PHQ-9 are not intended to be used in determining actual diagnoses due to limitations in their psychometric properties. For example, in case of the PHQ-9, the scoring method as well as the algorithm method were found to have good specificity but insufficient sensitivity for detecting cases of depression [71]. It is therefore not clear whether participants that were treated as cases of no depression in this study, were non-clinical cases. Consequently, the results must be interpreted taking into account that calculations on accuracy in this study are preliminary and based on a retrospective screening instrument.

Secondly, not all symptoms of depression were assessed at every single day during the 14-day assessment period. Even though it was required that symptoms were present on more than half of the assessment points, the ICD-10 criterion of the symptom being present for a minimum of 2 weeks on more than half the days was only approximated. While this may have reduced the validity of the screening result, it also significantly reduced the number of questions that participants had to answer from 45 to 9 questions per day. Given the relatively low number of missing values, the overall lower daily burden might have had a positive effect on retention.

Thirdly, since this was the first study on a new approach to depression screening there were no pre-existing findings to base cut-off criteria on. At this stage, the findings must be seen as preliminary and require replication–ideally in pre-registered trials.

Fourthly, as a side-effect of the question selection algorithm, a significant number of participants still had to be excluded due to taking up to 3 days longer to complete the minimum required number of AA assessments. At this stage, the algorithm was designed in this way to not frustrate users who are just missing a few questions for the generation of their 14-day report, but it was still decided to exclude these participants in order to match the assessment period with the PHQ-9. In future iterations of the algorithm, this can be avoided by limiting the Moodpath assessment period to 14 days while improving the question selection to optimize the compensation of missed question blocks in order to reduce the number of users with incomplete data after 14 days.

Fifthly, the study data were collected in a self-selected anonymous sample. The sample was not recruited to be a clinical sample and there is no information on participants' treatment or depression history or on potential physical reasons for reported symptoms. Different recruitment methods can result in systematic differences between samples of participants with depression symptoms [80]. Even though the PHQ-9 scores indicated a high symptom load, the sample needs to be treated as a non-representative population sample and furthermore needs to be contrasted with samples recruited in clinical settings in the future. Consequently, the generalizability of the results is limited. Furthermore, demographic data indicates that the sample was relatively young and mostly female, which further reduces generalizability to other demographic groups.

Finally, apps like Moodpath are constantly being improved based on research results and user feedback. Consequently, the screening algorithm and other elements of the app are likely to change in the future. Therefore, future versions of the app may work differently than the version 2.0.7 described in this paper.

## The Moodpath depression screening

The Moodpath version that was used in this study, utilized a simple screening algorithm that ensured that all 10 symptom categories of ICD-10 depression were assessed at least four times within a period of 14 days. If more than half of these assessments were above the cut-value of 2 on a 4-point severity scale, the symptom was counted as potentially clinically significant and was included in the Moodpath depression score. Given their 58% shared variance, the Moodpath depression score and the PHQ-9 score had significant overlaps but were still not found to be 100% congruent. Applying the ICD-10 criteria to the Moodpath data and comparing the result with the PHQ-9 algorithm (DSM-5 criteria), revealed a good screening accuracy when using a cut-value of 5. But here again, no perfect agreement was found (Table 3).

One potential explanation for the differences could be that both screenings did not measure exactly the same construct, given that Moodpath is based on the ICD-10 criteria for a depressive episode while the PHQ-9 assesses the DSM-5 criteria of Major depression. Despite the differences in symptom criteria, direct psychometric comparisons of the DSM and ICD are still rare. It was shown that both classification systems tend to be in agreement when it comes to severe or moderate cases of depression while the ICD-10 was found to be more sensitive in cases of mild depressive episodes [81]. However, since there are known issues with the sensitivity of the PHQ-9 [71], further investigations are needed to clarify whether the slightly higher number of cases identified by the Moodpath screening can be explained by a higher sensitivity of the smartphone-based screening.

Another possible explanation for the differences between both measures in our study are different influences of measurement error due to the momentary vs. the retrospective nature of the screenings. As has been discussed in the ambulatory assessment literature, retrospective questionnaires may be affected by several sources of bias, e.g. due to momentary mood at the time of assessment or recall bias [36–38]. The exploratory analyses in our study revealed that daily mood average and mood maximum explained a higher amount of PHQ-9 variance if only the day of the PHQ-9 assessment was considered. When considering earlier days, the explained variance was reduced and was lowest with the full 14-day dataset (for mood maximum) or 14 days before PHQ-9 assessment (for daily mood average). This may be an indicator of a momentary mood bias in answering the PHQ-9 questionnaire, as more positive mood at the time of answering the questionnaire may affect how participants retrospectively judge their symptoms. In line with this interpretation, mood average on the day of the PHQ-9 assessment explained additional PHQ-9 variance on top of the Moodpath depression score while 14-day average mood had no additional effect. Since short-term variations in momentary mood were common in our sample (see Supporting information), the PHQ-9 assessment may have been more severely affected by this type of bias while ambulatory measures are less likely to be systematically affected.

Since the Moodpath depression screening was a newly developed instrument, no validated cut-off scores exist yet. Consequently, the ICD-10 cut-off at 5 symptoms was used. Here, it is noteworthy that the same value was identified as optimum cut-off when not taking into account core and associated symptoms. Since the differences in sensitivity and specificity were marginal, there may be no advantage in differentiating between core and associated symptoms when using the ambulatory Moodpath data. Taking into account the overlapping confidence intervals and potentially different weightings of sensitivity and specificity, cut-off values at 4 or 6 can also be considered.

## Mood tracking

The exploratory results on the mood tracking component of the app suggest that ambulatory assessed mood dynamics may constitute a promising base for the prediction of depression

symptoms. In line with previous findings [57, 58], symptoms of depression measured with the PHQ-9 were associated with lower average mood, maximum and minimum mood, higher mood instability as well as higher mood variability. Our results specifically indicate that 14-day mood average has the strongest individual association with the severity of PHQ-9 depressive symptoms, which is still the case if only 3 to 4 days of data prior to PHQ-9 assessment are available. This finding illustrates that depression is an affective disorder that is mainly characterized by constant, negative mood states over a period of 14 days, as defined in the ICD-10. Another strong association was found for the mood minimum (i.e. maximally negative mood rating), even if there were only two or three assessments available from the day of the PHQ-9 assessment. While this underlines negative mood as a core symptom of depression, it is also important to note that the statistical association between the PHQ-9 score and mood indicators immediately prior to the PHQ-9 assessment may be confounded by a momentary mood bias. Other parameters for mood dynamics such as mood variability or mood instability were also found to be associated with depressive symptoms, but significantly less strongly and only with at least 8 days of data prior to the PHQ-9 assessment. The mood states of individuals with higher depression scores showed more variability and temporal instability (S2 and S3 Figs), which is in line with findings on the relation between affective instability and depression [53, 82]. In contrast to previous findings, mood variability and mood instability maintained their statistically significant association with depression symptoms when controlling for mood average. An interpretation of the data is that elevated affect instability or variability are indicators of trait neuroticism or negative affectivity [53]. These may not only be associated with current depression but could also be indicators for aspects of personality functioning, e.g. maladaptive emotion regulation, that are predictive of future depressive episodes [83, 84]. Emotion regulation difficulties seem to play a major role in the course of major depression [85]. Individuals with the same depression scores but different affect variability scores may therefore have a different course and prognosis of their depressive illness which could justify different treatment approaches.

Our analyses illustrate that combinations of several 14-day mood statistics may be a promising proxy for the ambulatory assessment of depression symptoms. If e.g. average 14-day mood and 14-day mood instability are combined, the model explains 49% of the PHQ-9 variance (vs. 57% explained by the 14-day Moodpath depression score alone). The results further indicate that, instead of replacing the Moodpath depression score with mood indicators, there may be an incremental value when adding mood indices on top of the symptom assessment. Consequently, the best performing model combined the symptom sum score with mood instability (62% explained PHQ-9 variance). This also indicates that ambulatory assessed mood ratings may not only comprise a proxy for depression symptom screening but may also add diagnostic value that is not covered by ambulatory assessed symptom ratings. Mood instability and mood variability added equal amounts of explained variance on top of the symptom ratings, underlining the value of information on the temporal dynamics of mood ratings. Still, it is important to note that the incremental value of affective dynamics in research on personality, well-being and psychopathology is increasingly being questioned by studies that aggregate datasets on indicators of affect dynamics [86–89]. In line with our findings, these studies identify mean level and general variability as the main indicators but find only little additional variance explained by other indicators of affect dynamics.

The analyses on mood statistics revealed several additional findings. For example, contrary to previous findings [53, 57, 58], low or high mood inertia did not significantly predict depression symptoms. This may be due to the larger time intervals between the mood ratings (i.e. 3 ratings per day compared to up to 10 ratings in previous studies). Furthermore, while there was a gender difference in the Moodpath depression scores (i.e., women having significantly

higher overall scores), no such effects were found on any of the mood dynamics parameters. This could be an indicator that direct mood measures are less susceptible to gender differences than ambulatory assessed depression symptom ratings.

## Burden of self-tracking

The Moodpath screening requires users to answer a set of three questions and one mood rating, three times a day for two weeks. While the scope of the ambulatory assessment is at the lower end of the continuum of what is common in research contexts, it is important to consider the burden on users and to balance it with the predictive value of additional data. Due to the recruitment procedures in this study (i.e., only including participants who completed the 14-day assessment), effects on retention cannot be estimated directly. However, since 178 out of 200 selected users agreed to answering the additional PHQ-9 questions after completing the 14-day assessment, there is indication that users who complete the assessment are still willing to continue with additional questions.

In the linear mixed effects model, a small but statistically significant time effect on mood ratings was found. Over the whole assessment period, mood ratings got slightly more negative. This could be an indication of negative emotional effects of self-tracking that are sometimes mentioned in the literature [66, 67]. However, data on this potential issue is still scarce and there are also findings that indicate that the frequent assessment of symptoms does not induce negative mood [90]. In our data as well, we did not find an effect of assessment frequency. Consequently, an alternative explanation for the found time effect on mood ratings could be symptom deterioration in the course of a depressive episode or a response bias. An unexpected initial elevation—especially of self-reported internal states—is a common phenomenon in studies with repeated measures. While it is usually stronger for negative states, it was also observed in assessments of positive states over time [91]. Consequently, there might have been a tendency of respondents to report higher mood scores at the beginning of the 14-day assessment period due to an initial elevation bias. To reach reliable conclusions on potential negative and positive effects of the screening, the Moodpath app needs to be tested in a randomized controlled trial. In addition, typical mood fluctuations over the course of a day, several days or even months and seasons should be considered in studies with larger sample sizes.

## Conclusions

This study found promising results in a preliminary evaluation of the Moodpath depression screening app. The associations found between Moodpath depression score and the PHQ-9 justify additional validation efforts. Furthermore, findings on the incremental value of mood statistics underline the importance of intensive repeated measurements with the potential for capturing individuals' mood dynamics across time. Since affective disorders are characterized by emotion regulation dysfunction and a certain degree of instability over time, the ambulatory assessment of affect and symptom states may therefore comprise information that are useful for treatment decisions beyond the classification of the respective affective disorder [92]. Due to methodological limitations, a core question remains: Is the Moodpath screening potentially more accurate than the PHQ-9 or is it the other way around? To further validate the screening approach, a comparison with a clinical interview such as the Structured Clinical Interview for DSM (SCID) is therefore essential.

As has been pointed out by Dubad et al. [69], the validation of ambulatory assessment tools on the basis of retrospective measures may not be the most robust validation approach due to fundamental differences in the cognitive processes that take place when participants generate an aggregated account of their symptoms versus a momentary account. The authors point out

that respondents answering a retrospective questionnaire may create an account that is based on additional emotional processing, e.g. taking context into account, and that this cannot necessarily be captured by ambulatory assessment methods. As a result, retrospective measures and momentary measures may as well be considered as incremental approaches in a multi-modal approach to depression screening.

In addition, ambulatory methods can be designed to capture a broader scope of relevant environment factors that may affect mood and symptom dynamics in situations and over time. This includes the potential to study interactions between affective dynamics and emotion regulation strategies in depression and other psychological disorders [59, 93]. Future improvements to the Moodpath algorithm could therefore aim at adding context information to the assessment (e.g., psychological situations) [94] to generate data sets that allow more complex statistical methods, such as dynamic network modelling, in order to gain more knowledge on interactions between context, mood and symptoms over time [44, 95]. Using a broader, population-based sampling approach, Moodpath could also be used to investigate potential early indicators of transition into depression such as critical slowing down in mood dynamics [63, 64].

The findings of this study can be used to development and improve smartphone-based screenings tools that further expand the potential of this assessment approach. Advanced apps have the potential to provide highly scalable and user-friendly screenings that aggregate different data sources such as ambulatory symptom ratings, momentary mood and the passive collection of sensor data over longer periods of time. Through interfaces with established diagnostic approaches and health care systems, smartphone apps have the potential to facilitate screening, monitoring and preventive measures. Furthermore, multimodal smartphone-based assessments may enable the identification of digital phenotypes, i.e. groups of people with distinguishable patterns of behavior assessed through smartphones sensors and interactions with the device [96]. In longitudinal study designs, digital phenotypes can be investigated in terms of their predictive validity with regard to clinical outcomes such as treatment response, treatment dropout, relapse rate, suicidality, quality of life or daily functioning. Future applications could be data-driven therapy-assistance systems that support decision making and enable precision mental healthcare [97].

## Supporting information

**S1 Fig. Individual mood trajectories over the 14-day assessment period (N = 113).** Assessment time = Exact time point of assessment; Mood rating = Momentary assessed mood score between 0 (negative mood) and 5 (positive mood).
(PDF)

**S2 Fig. Mood RMSSD for levels of depression severity (N = 113).** NS $p \geq .05$, $^{**}$ $p < .01$.
(PDF)

**S3 Fig. Mood SD for levels of depression severity (N = 113).** NS $p \geq .05$, $^{**}$ $p < .01$.
(PDF)

**S4 Fig. Mood average for levels of depression severity (N = 113).** $^{*}$ $p < .05$, $^{***}$ $p < .001$.
(PDF)

**S5 Fig. Mood inertia for levels of depression severity (N = 113).** NS $p \geq .05$.
(PDF)

**S1 Table. List of the 45 questions used for ambulatory assessment.**
(PDF)

**S2 Table. Order of the AA question blocks.**
(PDF)

## Author Contributions

**Conceptualization:** Sebastian Burchert, Johannes Zimmermann, Christine Knaevelsrud.

**Data curation:** Sebastian Burchert, André Kerber, Johannes Zimmermann.

**Formal analysis:** Sebastian Burchert, André Kerber, Johannes Zimmermann.

**Methodology:** Sebastian Burchert, André Kerber, Johannes Zimmermann, Christine Knaevelsrud.

**Supervision:** Johannes Zimmermann, Christine Knaevelsrud.

**Visualization:** Sebastian Burchert, André Kerber.

**Writing – original draft:** Sebastian Burchert, André Kerber, Johannes Zimmermann, Christine Knaevelsrud.

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
