## [Decision Letter · Decision Letter 0]

18 Nov 2019

PONE-D-19-24772

14-day smartphone ambulatory assessment of depression symptoms and mood dynamics in a general population sample: comparison with the PHQ-9 depression screening

PLOS ONE

Dear Dr. Burchert,

Thank you for submitting your manuscript to PLOS ONE. After careful consideration, we feel that it has merit but does not fully meet PLOS ONE’s publication criteria as it currently stands. Therefore, we invite you to submit a revised version of the manuscript that addresses the points raised during the review process.

We would appreciate receiving your revised manuscript by Jan 02 2020 11:59PM. To enhance the reproducibility of your results, we recommend that if applicable you deposit your laboratory protocols in protocols.io, where a protocol can be assigned its own identifier (DOI) such that it can be cited independently in the future. For instructions see: http://journals.plos.org/plosone/s/submission-guidelines#loc-laboratory-protocols

We look forward to receiving your revised manuscript.

Kind regards,

Gian Mauro Manzoni, Ph.D., Psy.D.

Academic Editor

PLOS ONE

Journal Requirements:

The authors declare that the research was conducted in the absence of any commercial or financial

relationships that could be construed as a potential conflict of interest. For the purpose of this study, Freie Universität Berlin and Aurora Health signed a formal data usage agreement. The data collection was conducted with regular users of the app. As part of the data usage agreement, Aurora Health provided the authors with a raw data export file from the app’s database. Aurora Health was not involved in the preparation of the raw data, the data analysis, the writing of this article and the interpretation the results.

Reviewers' comments:

Reviewer's Responses to Questions

**Comments to the Author**

1. Is the manuscript technically sound, and do the data support the conclusions?

Reviewer #1: Yes

Reviewer #2: Partly

2. Has the statistical analysis been performed appropriately and rigorously? 

Reviewer #1: Yes

Reviewer #2: I Don't Know

3. Have the authors made all data underlying the findings in their manuscript fully available?

Reviewer #1: Yes

Reviewer #2: No

4. Is the manuscript presented in an intelligible fashion and written in standard English?

Reviewer #1: Yes

Reviewer #2: Yes

5. Review Comments to the Author

Reviewer #1: This paper represents a pilot study in validating the mobile app for depression screening. For an initial pilot study, it was well-designed and used appropriate statistics. The authors were appropriately careful in the Discussion to point out the pilot nature and related limitations and future directions needed. The writing was clear and the logic was easy to follow. I have just a couple of suggestions for minor additions:

1) In terms of the Moodpath depression score, the authors acknowledge that they just chose a cutoff of 2 or higher on severity without empirical testing of whether that threshold was ideal. However, for the next criteria of the sum of all symptoms that had this “clinical significance on more than half of the ambulatory assessments” – the authors do not clarify how that criteria was chosen. Was that also just chosen without empirically determining whether ideal. For both, authors should clarify if these decisions for cut points were made a priori to inspection of the data.

2) Toward end of Discussion, the authors suggest that a structured clinical interview would be the ultimate criterion to use for future studies. I suggest they expand on that and note that longitudinal research is then needed to examine the predictive validity of important clinical outcomes such as response to treatment, relapse rate, subjective quality of life, quality of daily functioning, suicidal thoughts/attempts, etc. Ultimately, it would be the ability of the screening instrument to detect future behavior that would determine if it is "better" than a self-report measure that relies on retrospective recall. I agree that a first step for future research is to match to SCID diagnosis, but some discussion of further validation and potential use to predict best type of treatment and track changes during treatment would be useful too.

Reviewer #2: I have provided my detatiled comments in the uploaded document. In summary:

This paper reports on a diagnostic accuracy study which evaluates the ability of self‐reported ambulatory assessment of  depressive symptoms and mood monitoring to correctly classify participants as experiencing Major Depressive Disorder,  using the PHQ‐9 as the gold standard.  

This paper is timely as the use of ambulatory assessment and momentary monitoring of mood symptoms is gaining in use  ahead of the research evidence which should underpin it.  

The paper is fairly well written but it is long and could be shortened to improve clarity. The main concern I have is whether  there might be two papers presented as one here. Simply put, one is whether MOODPATH diagnosis agrees with a PHQ‐9  diagnosis and the other is whether mood monitoring agrees with the PHQ‐9 scores? It took me a number of times reading  through the manuscript to understand exactly what was being presented and whether they actually belonged together? I  just wonder if you are trying to achieve too much for one paper and clarity could be improved if you dealt with them  separately? Another concern is the large number of models presented and the lack of a pre‐published protocol. This needs  to be addressed.

6. PLOS authors have the option to publish the peer review history of their article (what does this mean?). If published, this will include your full peer review and any attached files.

Reviewer #1: Yes: Jeffrey S. Bedwell, Ph.D.

Reviewer #2: No

---

## [Author Response · Author response to Decision Letter 0]

29 Aug 2020

Please find our detailed responses in the attached document.

---

## [Decision Letter · Decision Letter 1]

23 Nov 2020

PONE-D-19-24772R1

Screening accuracy of a 14-day smartphone ambulatory assessment of depression symptoms and mood dynamics in a general population sample: comparison with the PHQ-9 depression screening

PLOS ONE

Dear Dr. Burchert,

Thank you for submitting your manuscript to PLOS ONE. After careful consideration, we feel that it has merit but some minor suggestions to further improve the manuscript remain. Therefore, we invite you to submit a revised version of the manuscript that addresses the points raised during the review process.

The reviewers indicated that all former comments have been addressed. However, a new reviewer suggested some minor revisions to further improve the manuscript, and I particularly agree on the need of more clarity about the recruitment pocess and that a Flow-chart would help to address this issue.

We look forward to receiving your revised manuscript.

Kind regards,

Ricarda Nater-Mewes, PhD

Academic Editor

PLOS ONE

Additional Editor Comments (if provided):

The reviewers indicated that all former comments have been addressed. However, a new reviewer suggested some minor revisions to further improve the manuscript, and I particularly agree on the need of more clarity about the recruitment pocess and that a Flow-chart would help to address this issue.

Reviewers' comments:

Reviewer's Responses to Questions

**Comments to the Author**

1. If the authors have adequately addressed your comments raised in a previous round of review and you feel that this manuscript is now acceptable for publication, you may indicate that here to bypass the “Comments to the Author” section, enter your conflict of interest statement in the “Confidential to Editor” section, and submit your "Accept" recommendation.

Reviewer #1: All comments have been addressed

Reviewer #3: All comments have been addressed

2. Is the manuscript technically sound, and do the data support the conclusions?

Reviewer #1: Yes

Reviewer #3: Yes

3. Has the statistical analysis been performed appropriately and rigorously? 

Reviewer #1: Yes

Reviewer #3: Yes

4. Have the authors made all data underlying the findings in their manuscript fully available?

Reviewer #1: Yes

Reviewer #3: Yes

5. Is the manuscript presented in an intelligible fashion and written in standard English?

Reviewer #1: Yes

Reviewer #3: Yes

6. Review Comments to the Author

Reviewer #1: The authors were able to address my previous concerns and suggestions. It also appears that they have responded to all of the points from the other reviewer. I have no further suggestions for changes.

Reviewer #3: This is an important, novel paper which goes some way to addressing the kinds of questions associated with the surfeit of mobile apps available to people with mental health difficulties and their clinicians. The paper has described clearly and accurately the literature and does a mostly thorough job in doing so.

The study itself appears to be appropriate in terms of design and statistical analysis and the conclusions are balanced and realistic. I believe that the previous reviews have been addressed satisfactorily. I did feel there were some issues around the clarity of the methodology some of which could probably be addressed with figures rather than text.

A few comments:

Line 75: "The willingness to seek mental health information online was found to be higher among

persons who experience increased psychological distress" - Higher compared to whom?

Line 85 - poorly constructed sentence. Remove 'for' and consider changing the word obscure for something other such as opaque, complex or hard to navigate

Page5 - What are the figures for health or other app use in the general population - how does this compare to mental health app use?

Page 8 - summary of pilot study aims could be more clear for the reader - strip back to the basics here.

Page 9 - more clarity needed over recruitment and when and how were onboarded - a diagram here would be helpful to explain the process. For example its not immediately apparent when participants start the study. If they are already users - does this impact their results and feedback? Do new users show greater effect from a usage and depression point of view?

Page 10 - the app has already received a CE - does this mean the app has already undergone validation testing for safety and effectiveness?

Page 16 - Is the correction type for multiple testing enough here?

Page 18 and conclusions - Respondents were more likely to complete the tracker in the evening - how did this affect mood - changes in hormone levels, diurnal variance etc. should be considered here perhaps? could this be stratified?

7. PLOS authors have the option to publish the peer review history of their article (what does this mean?). If published, this will include your full peer review and any attached files.

Reviewer #1: **Yes: **Jeffrey Bedwell

Reviewer #3: No

---

## [Author Response · Author response to Decision Letter 1]

18 Dec 2020

-----

Response to editor

-----

Additional Editor Comments:

The reviewers indicated that all former comments have been addressed. However, a new reviewer suggested some minor revisions to further improve the manuscript, and I particularly agree on the need of more clarity about the recruitment process and that a Flow-chart would help to address this issue.

Thank you very much! Please find our replies to reviewer 3 below.

We also used this opportunity to add the following paragraph in the discussion on the latest findings regarding the limited incremental value of affective dynamics in research on personality, well-being and psychopathology (see line 675):

“Still, it is important to note that the incremental value of affective dynamics in research on personality, well-being and psychopathology is increasingly being questioned by studies that aggregate datasets on indicators of affect dynamics [86-89]. In line with our findings, these studies identify mean level and general variability as the main indicators but find only little additional variance explained by other indicators of affect dynamics.” 

-----

Response to reviewers

-----

Reviewers' comments:

Reviewer #3: This is an important, novel paper which goes some way to addressing the kinds of questions associated with the surfeit of mobile apps available to people with mental health difficulties and their clinicians. The paper has described clearly and accurately the literature and does a mostly thorough job in doing so.

The study itself appears to be appropriate in terms of design and statistical analysis and the conclusions are balanced and realistic. I believe that the previous reviews have been addressed satisfactorily. I did feel there were some issues around the clarity of the methodology some of which could probably be addressed with figures rather than text.

A few comments:

1) Line 75: "The willingness to seek mental health information online was found to be higher among persons who experience increased psychological distress" - Higher compared to whom?

Reply: Thank you for pointing out this missing piece of information. We adjusted the sentence accordingly: 

“Among internet users in a random sample of general practice patients in Oxfordshire (England), the willingness to seek mental health information online was found to be higher among persons who experience increased psychological distress and even higher in persons with a past history of mental health problems [17].”

2) Line 85 - poorly constructed sentence. Remove 'for' and consider changing the word obscure for something other such as opaque, complex or hard to navigate

Reply: We agree that the wording in this sentence is not ideal and have adjusted the sentence as follows:

“However, the number of freely available mental health apps has long become opaque for clinicians and end users seeking trustworthy, secure and evidence-based apps [21, 27, 28].”

3) Page5 - What are the figures for health or other app use in the general population - how does this compare to mental health app use?

Reply: To our knowledge, there are no population-based studies on mental health app use in the general population of a country. However, it can be assumed that – compared to the widespread use of smartphones (up to 90% of the population, depending on the country) – mental health app use is still relatively rare. In a German population-based study (Ernsting et al., 2017), 20,54% of smartphone owners had used health related apps (most commonly for smoking cessation, weight loss or a healthy diet). Mental health apps were not covered separately, but Baumel et al. (2019) showed that there is currently only a very small number of mental health related apps that achieved more than 100.000 total downloads (worldwide). In addition, the authors showed that most of these apps struggle with very low adherence rates. At the same time, mental health apps seem to become more relevant for those who experience psychological distress or who are already suffering from a psychological disorder. To keep the introduction concise, we included findings on this (see page 4) but did not go into more detail on the use of (health) apps in the general population. 

4) Page 8 - summary of pilot study aims could be more clear for the reader - strip back to the basics here.

Reply: We shortened this section and reduced the level of detail to make it easier for the reader to identify the main aims of the study.

5) Page 9 - more clarity needed over recruitment and when and how were onboarded - a diagram here would be helpful to explain the process. For example its not immediately apparent when participants start the study. If they are already users - does this impact their results and feedback? Do new users show greater effect from a usage and depression point of view?

Reply: We agree that this requires more clarity because it is very relevant to understand that the participants in our study were already regular users of the app before they were asked to participate in the study. This applies to all participants in the study. Therefore, we added the following explanation on page 9:

“Participants in this study were regular users of the app who had started using the app before being asked to participate in the study.”

Consequently, the in-app data was initially not gathered in a study context but in a regular app use context and participants retrospectively provided consent for using this data in the study. To further illustrate this, we have expanded Fig 2 by adding information explaining that – prior to the study – participants were regular users who completed the app’s ambulatory assessment phase. These users were then contacted automatically through an in-app notification and asked to participate.

Only data of app users who provided consent to participate in this study is included in the paper. Therefore, we cannot provide descriptive statistics on the user flow prior to giving consent.

Since we do not have a comparison group, it is not possible to say whether regular and study users differ in usage or symptom patterns. All participants were regular app users before participating in the study by answering the PHQ-9 items and we therefore assume that the findings might have a higher ecological validity compared to a different study design with sample of non-app users who are invited to install the app for the study. 

6) Page 10 - the app has already received a CE - does this mean the app has already undergone validation testing for safety and effectiveness?

Reply: In 2017, the manufacturer of the app received a CE medical device certificate as a Class I medical device. To our knowledge, at the time this did not require pre-existing validation or effectiveness studies on the product itself. Additional studies on the app have been and are being conducted. But to our knowledge these studies are not published yet. Our study is the first one to investigate the psychometric properties of the app’s screening approach. 

7) Page 16 - Is the correction type for multiple testing enough here?

Reply: We think the approach is sufficient in this case. A total of 155 tests are being conducted as part of the explorative analysis. At the chosen p < .01 we can expect that approximately two tests are false positives.

8) Page 18 and conclusions - Respondents were more likely to complete the tracker in the evening - how did this affect mood - changes in hormone levels, diurnal variance etc. should be considered here perhaps? could this be stratified?

Reply: This could result in a bias if mood scores at the evening assessments differed significantly from mood scores at midday or evening. We tested this and did not find statistically significant differences between the three assessment periods in the data. Therefore, we did not apply measures to control for this type of bias and did not include potential explanations for daytime specific variation on mood ratings in the discussion.

---

## [Editor Report · Decision Letter 2]

21 Dec 2020

Screening accuracy of a 14-day smartphone ambulatory assessment of depression symptoms and mood dynamics in a general population sample: comparison with the PHQ-9 depression screening

PONE-D-19-24772R2

Dear Dr. Burchert,

We’re pleased to inform you that your manuscript has been judged scientifically suitable for publication and will be formally accepted for publication once it meets all outstanding technical requirements.

Kind regards,

Ricarda Nater-Mewes, PhD

Academic Editor

PLOS ONE

Additional Editor Comments (optional):

Thank you for the revised version of your manuscript. All suggestions by the reviewer have been addressed thoroughly, and I consider the manuscript suitable for being published.
---

## [Editor Report · Acceptance letter]

26 Dec 2020

PONE-D-19-24772R2 

Screening accuracy of a 14-day smartphone ambulatory assessment of depression symptoms and mood dynamics in a general population sample: comparison with the PHQ-9 depression screening 

Dear Dr. Burchert:

I'm pleased to inform you that your manuscript has been deemed suitable for publication in PLOS ONE. Congratulations! Your manuscript is now with our production department. 

Kind regards, 

on behalf of

Dr. Ricarda Nater-Mewes 

Academic Editor

PLOS ONE